# Description and Analysis of Horse Swimming Strategies in a U-Shaped Pool

**DOI:** 10.3390/ani15020195

**Published:** 2025-01-13

**Authors:** Pauline Gaulmin, Frédéric Marin, Claire Moiroud, Audrey Beaumont, Sandrine Jacquet, Emeline De Azevedo, Pauline Martin, Fabrice Audigié, Henry Chateau, Chloé Giraudet

**Affiliations:** 1Centre d’Imagerie et de Recherche sur les Affections Locomotrices Equines (CIRALE), Ecole Nationale Vétérinaire d’Alfort, 94700 Maisons-Alfort, France; pauline.gaulmin@vet-alfort.fr (P.G.); claire.moiroud@vet-alfort.fr (C.M.); audrey.beaumont@vet-alfort.fr (A.B.); sandrine.jacquet@vet-alfort.fr (S.J.); emeline.de-azevedo@vet-alfort.fr (E.D.A.); fabrice.audigie@vet-alfort.fr (F.A.); 2Laboratoire de BioMécanique et BioIngénierie (UMR CNRS 7338), Centre of Excellence for Human and Animal Movement Biomechanics (CoEMoB), Université de Technologie de Compiègne (UTC), Alliance Sorbonne Université, 60200 Compiègne, France; frederic.marin@utc.fr (F.M.); chloe.giraudet@utc.fr (C.G.); 3LIM France, 24300 Nontron, France; pmartin@lim-group.com

**Keywords:** horse swimming, swimming cycle, equine locomotion, rehabilitation, motion capture

## Abstract

Swimming is often used in horse training and rehabilitation, but there has been little detailed analysis of how horses coordinate their limbs in water. We first conducted a qualitative analysis where expert veterinarians visually identified the propulsion and return phases, allowing for an initial characterization of swimming patterns. Then, we used a computer model to classify the swimming patterns based on specific measurements. The model agreed with the veterinarians’ observations 96.8% of the time. We found that all horses followed a common movement pattern but showed enough differences to identify three swimming styles: a two-beat strategy with side-to-side overlap, a two-beat strategy with diagonal overlap, and a four-beat strategy where each limb moved independently. These findings open up new possibilities for research, including tracking changes in swimming styles over time and exploring how swimming can help with different injuries. This work will help improve aquatic training programs for horses.

## 1. Introduction

Aquatic training of horses in pools has been used for many years [1]. The primary motivation is to preserve the physical condition of horses [2] through a significant level of activity, such as swimming, while minimizing the stress on the limbs associated with weight-bearing forces [3] thanks to the buoyancy provided by immersion in a pool. This principle has also been applied in rehabilitation [4] to promote early weight bearing. The buoyancy of water reduces the impact and stress on vulnerable structures, allowing for earlier engagement in weight-bearing activities compared to rehabilitation programs without swimming [5]. This approach has the potential to accelerate recovery, maintain muscle strength and joint mobility, and prevent the adverse effects associated with prolonged immobilization [6]. In 1981, Smith successfully demonstrated the effectiveness of this approach on six horses with one or more fractures, reducing stress on healing bones [4]. The quantification of weight reduction is based on Archimedes’ principle. According to this principle, a body immersed in a fluid experiences an upward thrust equal to the weight of the fluid displaced.

However, horses remain poorly adapted to an aquatic environment [7]. Due to their morphology, there is an uneven distribution of mass in water; denser parts such as the limbs, head, and neck tend to sink, while less-dense regions, such as the thorax, tend to float. This imbalance creates a righting moment that, if the animal remains passive, gradually returns it to an equilibrium position that is nearly vertical [8]. Despite this, the horse instinctively begins to swim as soon as its hooves can no longer touch the ground. Immersion places the horse in an unfamiliar environment, governed by different physical laws [3], particularly those related to hydrodynamic resistance and buoyancy [3]. These constraints compel the horse to alter its usual locomotor behavior, leading to increased range of motion [9,10,11] and heightened muscular activity [12]. The increased resistance to limb movement in water reactivates agonist muscles, reduces co-contraction of paired antagonist muscles, and enhances neuromuscular control and muscle activity coordination [3].

Physiological adaptations also occur during aquatic training in horses. Numerous studies have focused on the horse’s physiologic responses to effort in the pool, beginning with Murakami in 1976 [1] and followed in the 1980s by Nicholl and Fregin [13], and Irwin and Howell [2]. These adaptations affect various body systems and contribute to the overall benefits of this training method. Notable adaptations include changes in cardiovascular function [1,13,14], mechanics of breathing [15,16,17], and metabolic processes [1,14], all of which enhance the horse’s efficiency in the aquatic environment and may potentially lead to improve performance on land as well [18]. In addition to these physiological changes, the locomotor pattern is obviously also modified during swimming.

Although swimming has been increasingly incorporated into training programs by many trainers on an empirical basis, very few studies have focused on the biomechanical analysis of horses swimming in a pool, particularly concerning limb coordination [9,10,19]. The first publications on aquatic work for horses were reported by Swanstrom and Lindy in 1973 [20], where they described a 3-beat coordination pattern during horse swimming for the first time. However, they also observed variations in coordination patterns. For some horses, the pattern resembled that of a terrestrial walk, while for others, it was closer to a terrestrial trot. These observations were made in a circular pool where the act of turning imposes additional constraints on movement. In 2024, a study conducted in a semi-circular pool identified three distinct swimming patterns, categorizing them as connected pacer, disconnected pacer, and rear engine [19]. However, these observations were qualitative. The first kinematic analysis of limb movement during swimming was presented in 2021 [9,10]. These studies focused on the rotation in the sagittal plane of the front limb joints (elbow, carpus, and fetlock) [9] and of the hindlimb joints (stifle, hock, and fetlock) [10]. This work provided an initial understanding of the expected range of motion for different joints during swimming. However, these studies did not fully describe the swimming cycle and independently analyzed the kinematics of the front limbs and hindlimbs. In 2023, Giraudet et al. [11] furthered this work by developing a marker-based motion capture system that enabled 3D motion capture of the horse throughout a complete swimming cycle. This work also allowed for the determination of joint angles used during swimming, with an accuracy of 1°. The 3D angles of the elbow and stifle joints served as the basis for defining a swimming cycle by identifying two key events corresponding to the moments of propulsion and return, in analogy with terrestrial events [21].

In terrestrial locomotion, a stride is divided into two main phases: the stance phase, where the hoof is in contact with the ground, supporting the body weight while moving backward (retraction), and the swing phase, where the hoof moves forward without ground contact (protraction) [22]. Each of these phases can be further broken into three stages. The stance phase includes shock absorption (closing of joint angles upon ground contact), support (when the limb is vertical relative to the ground), and propulsion (opening of joint angles during rapid limb retraction). The swing phase comprises return (closing of joint angles as the limb leaves the ground), suspension (intermediate stage where joints are maximally flexed), and ground coverage (opening of joint angles ending with ground contact) [22]. The main difference during swimming lies in the absence of ground contact, which alters the analogy of stride phases. The propulsion phase in the pool cannot be restricted to the same definition as the propulsion phase in terrestrial locomotion, as propulsion occurs earlier in the water due to the absence of a shock absorption phase. The equivalent of the terrestrial swing phase is termed “return”, during which the limb moves forward again. Additionally, an extra phase exists in the pool, which can be considered as a “sliding phase”, occurring between the propulsion and return phases. This phase corresponds to a period where the stationary limb benefits from the inertia of movement in water. In summary, during swimming, the propulsion phase encompasses the movement of a limb from the start of its backward motion until it stops at the end of that backward motion, corresponding to the retraction movement. The return phase, on the other hand, encompasses the limb’s movement from the beginning of its forward motion until it stops at the end of that motion, corresponding to the protraction movement [23].

Studying limb coordination during a swimming cycle is essential for understanding the biomechanics of aquatic movement. To analyze this coordination, we drew inspiration from established methods in terrestrial biomechanics, creating a link between these two fields of study. In terrestrial biomechanics, gaits are analyzed and classified according to their linear, temporal, and dynamic characteristics. This approach has led to the development of analytical tools like coordination graphs, which visually represent the sequence of limb movements as a function of the percentage of stance phase duration during a stride [23]. By applying these methods to swimming, we can meticulously describe limb movements in water, compare aquatic and terrestrial coordination patterns, and identify adaptations specific to the aquatic environment. In the pool, the bar corresponding to the terrestrial stance phase will be replaced by the propulsion phase. Using these coordination graphs, various indicators can be calculated for a quantitative analysis of gaits in terrestrial biomechanics, such as the duty factor (the ratio of the stance phase to the stride duration), overlap (the period during which two or more specified limbs are simultaneously in stance), and advance placement (the time interval between the ground contact of two specified limbs) [23]. Although these indicators were originally developed for terrestrial analysis, they can be adapted for studying swimming biomechanics.

The aims of this study are twofold: first, to determine the coordination pattern of the limbs during swimming, and second, to identify quantitative parameters based on spatio-temporal events that can describe and distinguish its variations. The first objective was achieved through the visual detection of propulsion and return phases by expert veterinarians, followed by a qualitative analysis of the different swimming patterns. The second objective involved classifying the swimming laps using a machine learning algorithm, based on two spatio-temporal ratios, α and β. The hypothesis was that specific thresholds for α and β would allow for a more rational classification of swimming strategies, thereby improving their description and understanding.

## 2. Materials and Methods

### 2.1. Population and Protocol

This study was conducted in a U-shaped pool, 50 m long and 3 m deep, located at CIRALE (EnvA, Goustranville, France). The water temperature ranged between 15 and 22 °C depending on the season. Eleven non-lame amateur show jumping horses, without limb injuries but potentially exhibiting cervical or dorsal lesions compatible with pool exercise, were included in this study (see Appendix A for details of the inclusion process and excluded horses). Only one horse had already swum before, but all of them benefited from three familiarization sessions in a pool before inclusion. The main characteristics of the horses included in this study are summarized in Table 1. Horses #04, #05, #07, and #10 were excluded from the study prior to the 8th week of training, either for clinical (onset of lameness, colic) or practical reasons (allergy to the glue used in the experiment, for example).

A standardized twelve-week training program was implemented uniformly for all of the horses. The training began with a 4-week period of land-based exercises only (flatwork and jumping) to standardize the exercise level of all the horses at the start of the protocol. In the fifth week, the horses transitioned from solely land-based exercises to a combination of land-based and water-based exercises, incorporating three swimming sessions per week. A swimming session consisted of laps, which included entering the pool, swimming in the U-shape pool, and exiting the pool (Figure 1). The horses were deliberately allowed to swim at their self-selected speeds to ensure natural and undisturbed swimming behavior during data collection. After a three-week adaptation period (weeks 5 to 7, with 9 swimming sessions, during which the horses wore monitoring equipment), the measurements were taken during the 8th week of training. This swimming session included two warm-up laps with one to the left and one to the right, followed by ten laps in the pool, five laps to the left and five to the right, interspersed with a walking break on a treadmill (Figure 1). This swimming session structure applied to all of the horses except for horses #01 and #02. After the warm-up, these two horses performed three laps to the right, took a break on the treadmill, then completed three laps to the left, followed by another break on the treadmill, and finally finished with three laps to the right.

### 2.2. Video Acquisition, Definition, and Detection of the Events Related to the Swimming

Underwater video recordings were performed using 6 synchronized GoPro Hero 8 cameras (GoPro Inc., San Mateo CA, USA), operating at 120 frames per second with a 2.7 K resolution [11]. This setup provided a 3-m-long field of view, allowing for the visualization of at least one complete swimming cycle (Figure 2).

Aquatic cycling was defined by analogy with the terrestrial definition of a stride. It corresponded to a complete cycle of repetitive limb movements, ending when the reference limb has returned to its starting position. In the aquatic environment, the propulsion phase (PP) was defined as the backward movement of the limb until it stopped moving backward. Conversely, the return phase (RP) corresponded to the forward movement of the limb until it stopped moving forward. Thus, the beginning of the propulsion phase marked the end of the return phase, and the end of the propulsion phase corresponded to the beginning of the return phase (Figure 3).

These events were tagged on the video based on the joint angles formed by the most proximal visible segments, according to the method and results of Moiroud et al. [21].

### 2.3. Sequence of the Limb Movements During Swimming and Coordination Diagram

First, a qualitative analysis of the sequence of the limbs was performed visually by two experienced veterinary specialists in equine locomotion using the videos [24]. The analysis of the propulsion phases allowed for the identification of the following swimming sequence, which begins with (1) the left forelimb, followed by (2) the left hindlimb, then (3) the right forelimb, and finally (4) the right hindlimb. Additionally, by observing the timing of the propulsion phases, the veterinary team qualitatively identified three different swimming strategies (Figure 4). Strategy 1 (S1) was a two-beat swimming strategy with lateral timing overlap, characterized by the pairing of propulsion phases of limbs in lateral pairs, fore and hind. This allowed for the identification of overlaps between the left forelimb and left hindlimb as well as the right forelimb and right hindlimb. Strategy 2 (S2) was a two-beat swimming strategy with diagonal timing overlap, based on the pairing of the propulsion phases of the limbs in diagonal pairs, both fore and hind. This resulted in overlaps between the left hindlimb and right forelimb, as well as the right hindlimb and left forelimb. Finally, Strategy 3 (S3) was a four-beat swimming strategy, where the movements of each limb were executed independently in a sequential manner. This strategy exhibited variations that were noticeable to the naked eye, suggesting the potential existence of subcategories, although these were difficult to clearly distinguish through visual observation alone.

Second, a quantitative analysis was performed as follows. Using the tagged video data for the propulsion and return timings of each limb, a coordination diagram was generated, representing the percentage of each phase of the swimming cycle (Figure 5a).

From these diagrams, quantitative parameters have been defined as follows (Figure 5b):Duty factor (DF): This parameter expresses the duration of the propulsion phase for a specified limb as a percentage of the total cycle duration ($T_{cycle}$). It is calculated as $DF=\frac{PP}{T_{cycle}}$, where PP is the propulsion phase. The duty factors for individual limbs are noted as $DF_{LF}$ for the left forelimb, $DF_{RF}$ for the right forelimb, $DF_{LH}$ for the left hindlimb, and $DF_{RH}$ for the right hindlimb. $DF_{Front}$ is defined as the average of $DF_{LF}$ and $DF_{RF}$, while $DF_{Hind}$ is the average of $DF_{LH}$ and $DF_{RH}$.Overlap (OL): Overlap has been defined as the percentage of the cycle during which two specified limbs are simultaneously in the propulsion phase. Lateral overlap (latOL) examines two ipsilateral fore and hindlimbs, with $latOL_{left}$ for the left limbs and $latOL_{right}$ for the right limbs. Diagonal overlap (DiagOL) examines two diagonal fore and hindlimbs, with $DiagOL_{left}$ for the left (LF-RH) and $DiagOL_{right}$ for the right diagonal (RF-LH).Sliding (SL) phase: This phase (expressed as a percentage of the cycle duration) was defined when the propulsion of two consecutive limbs in the sequence does not overlap. Lateral sliding (latSL) examines two ipsilateral fore and hindlimbs ($latSL_{left}$ for the left limbs and $latSL_{right}$ for the right limbs), while diagonal sliding (DiagSL) examines two diagonal fore and hindlimbs ($DiagSL_{left}$ for the left and $DiagSL_{right}$ for the right diagonal). For computational purposes, a positive sliding phase will be considered as a negative overlap. For example, when $latSL_{left}>0$, $latOL_{left}=-latSL_{left}$.Delay (DL): delay was defined as the percentage of the cycle elapsed between the beginning of propulsion of one specified limb and the beginning of propulsion of a second specified limb. Diagonal delay (DiagDL) measures the delay between a diagonal pair ($DiagDL_{left}$ for the left diagonal and $DiagDL_{right}$ for the right diagonal). Lateral delay measures the delay between two ipsilateral limbs ($LatDL_{left}$ for the left side and $LatDL_{right}$ for the right side). Contralateral delay measures the delay between two contralateral forelimbs or hindlimbs ($CoLatDL_{fore}$ for the forelimbs and $CoLatDL_{hind}$ for the hindlimbs). The mean lateral delay (MeanLatDL) is then defined as the average of $LatDL_{left}$ and $LatDL_{right}$, and the mean diagonal delay (MeanDiagDL) is the average of $DiagDL_{left}$ and $DiagDL_{right}$.

Additional composite parameters α and β were calculated as follows: (1)$$\alpha =\frac{MeanLatDL}{DF_{Front}},$$(2)$$\beta =\frac{MeanDiagDL}{DF_{Hind}},$$

The lateral ratio α (Equation (Equation 1)) assesses the relative percentage of delay of a hindlimb and the corresponding lateral forelimb. As α decreases, the synchronization between the forelimb and hindlimb on the same side (RF-RH or LF-LH) increases. The diagonal ratio β (Equation (Equation 2)) measures the relative percentage of delay between a forelimb and the corresponding diagonal hindlimb. Similarly, as β decreases, the synchronization between the limbs on the same diagonal (LH-RF or RH-LF) increases.

To validate the discrimination power of the composite parameters, a classification was performed via two-step clustering using k-means in Python 3.9.13 (scikit-learn python library, Python Software Foundation, Beaverton, OR, USA) (Figure 6). With the number of clusters set to 2, the first classifier segregated high versus low α values to distinguish the laps with a low lateral ratio from others by a threshold $\alpha_{1}$ and isolate a category A (CatA) when $\alpha \leq \alpha_{1}$. With the number of clusters set to 3, the second classifier separated 3 other categories according to the diagonal ratio β via the determination of two thresholds, $\beta_{1}$ and $\beta_{2}$. Category B (CatB) was identified when $\beta <\beta_{1}$, Category C (CatC) when $\beta_{1}\leq \beta <\beta_{2}$, and Category D (CatD) when $\beta_{2}\leq \beta$. At the end of the classification process, four categories (A to D) were obtained, whereas only three strategies had been identified by the veterinarians. The fourth category was introduced to account for at least two subcategories, reflecting the qualitative observation that the four-beat strategy (S3) showed variations.

The correspondence between the strategies identified by the veterinarians and the categories derived from the classification procedure was evaluated using a confusion matrix and the classical associated metrics. The overall accuracy (the number of correctly classified laps over the total number of laps) was then reported for the two-steps classifier. The principle of category-wise classifiers [25] will be used to give an overview of the results of the two-step classifier on each category. For each category-wise classifier, precision (considering a category Cat, the precision is the number of correctly labeled laps divided by the number of laps labeled as Cat by the two-step classifier, $precision=\frac{number\_of\_laps\_in\_Cat\_labeled\_as\_Cat}{number\_of\_laps\_labeled\_as\_Cat}$), recall (the recall is the number of correctly labeled laps divided by the number of laps in the category considered, $recall=\frac{number\_of\_laps\_in\_Cat\_labeled\_as\_Cat}{number\_of\_laps\_in\_Cat}$), and F1 score (the harmonic mean of precision and recall, $F1-score=2\frac{precision*recall}{precision+recall}$) were computed.

Finally, the values of the quantitative parameters were presented for each category. Due to our data’s varied distributions and differing sample sizes across categories (ranging from a number to its double), we opted to perform permutation tests on each parameter with all category pairs to analyze statistically the differences in the spatio-temporal parameters within categories (the number of permutations is set to 10,000, while the *p*-value is 0.001).

## 3. Results

### 3.1. Swimming Strategies Identified Visually by Expert Veterinarians

The results of the identification of the different swimming strategies by the veterinarians are compiled in Table 2.

### 3.2. Classification into Categories Using the Two-Step Classifier and Reference Thresholds

The clustering in the plane (α,β) performed by the two-step classifier allowed for the identification of the four categories {CatA, CatB, CatC, CatD} (Figure 7). This clustering allowed us to define the values of the three thresholds. The threshold $\alpha_{1}=0.58$ of the lateral ratio separated CatA from the three others, while the values of the diagonal ratio $\beta_{1}=0.63$ and $\beta_{2}=0.91$ divided the right part of the plane to segregate CatB, CatC, and CatD.

The results of the identification of the different swimming categories by the two-step classifier are compiled in Table 3.

### 3.3. Comparison Between Visually Based Segregation (Strategies) and Computer-Based Segregation (Categories) of the Swimming Laps

The confusion matrix between classification categories based on the clustering in the plane (α,β) and the visually identified strategies is presented in Figure 8.

The confusion matrix gave an overall accuracy of 96.8%. The category-wise results are given by Table 4. The precision was over 0.91 for all categories, with the most false positives for CatB (2 laps of S3 were mislabeled as CatB by the two-step classifier). The recall was greater than 0.96 for all categories, with the most false negatives for CatC + D (3 laps of S3 were mislabeled, 1 in CatA and 2 in CatB) and the best case for CatB. Finally, the F1 score was then above 0.95 for all categories. Data analysis allowed us to show the match between the visually identified strategies and the algorithmic classified categories. These results allowed us then to deduce that category CatA corresponded to strategy S1 (two beats with mainly lateral overlap), category CatB to strategy S2 (two beats with mainly diagonal overlap), and categories CatC and CatD to different variations of strategy S3 (four beats), with CatC comprising four distinct beats with a medium percentage of overlap, and CatD comprising four distinct beats with a low percentage of overlap.

### 3.4. Description and Examples of Coordination Graphs for Each Category

In the plane (α,β), the threshold $\alpha_{1}=0.58$ separated the lateral two-beat swimming identified by the veterinarians as strategy S1 from the other strategies. Below this threshold $\alpha_{1}$, the horse was lateralized: the propulsion movement of the hindlimb started close to the propulsion movement of the forelimb on the same side (Figure 9a).

The threshold $\beta_{1}=0.63$ separated, within the other strategies, the diagonal two-beat coordination pattern identified by the veterinarians as S2 from the four-beat ones (S3). Below this threshold, the horse was diagonalized: the propulsion movement of the forelimb started close to the propulsion movement of the hindlimb located diagonally (Figure 9b).

The threshold $\beta_{2}=0.91$ separated the four-beat coordination pattern identified by the veterinarians as strategy S3 among themselves. If the horse fell between the $\beta_{1}$ and $\beta_{2}$ thresholds, it swam with four distinct beats with a medium percentage of overlap (Figure 9c). Above this threshold, the horse swam with four distinct beats and a low percentage of overlap (Figure 9d). Inside this last case, the horse had an intermediate coordination pattern, which was neither really lateralized nor really diagonalized, but it swam with four distinct beats with different percentages of overlaps: the propulsion movement of the forelimb started more or less far from the propulsion movement of the hindlimb. This would create both lateral and diagonal overlaps, but in equivalent proportions.

### 3.5. Inter- and Intra-Individual Variability in the Swimming Patterns

Out of the 125 laps, 49.6% were performed using a four-beat swimming category and 50.4% were performed using a two-beat swimming category. A lateral two-beat category (catA) was observed in 32.8% of the recorded laps, a diagonal two-beat category (catB) was observed in 17.6% of the recorded laps, a four-beat category with a medium overlap percentage (catC) was observed in 34.4% of the recorded laps, and a four-beat category with a low overlap percentage (catD) was observed in 15.2% of the recorded laps. A total of 49.6% of the laps were performed on the left, and 50.4% were performed on the right. Among the left laps, 28.8% were performed using a four-beat category and 20.8% were performed using a two-beat category (15.2% observed in catA and 5.6% observed in catB). As for the right laps, 20.8% were performed using a four-beat category and 29.6% were performed using a two-beat category (17.6% observed in catA and 12.0% observed in catB).

Five horses maintained the same strategy throughout the laps: horse #08 and horse #11 used a lateral two-beat category (catA), horse #01 adopted a diagonal two-beat category (catB), while horses #02 and #06 employed a four-beat category. Six horses changed their category between laps. These horses used different combinations of categories, including a combination of four-beat and lateral two-beat for horses #03, #12, #13, and #14, as well as a combination of three categories for horses #09 and #15 with a different proportion of category uses. For horses employing two categories, the dominant category was identical and corresponded to a four-beat category. However, for horses employing three categories, the dominant category was different and corresponded to a four-beat category for horse #09 and a two-beat diagonal overlap category (catB) for horse #15.

The intra-individual results presented in Appendix B showed that four horses changed category after a break, specifically horses #03, #06, #11, and #13. Four horses also changed category after a treadmill session and changing direction, specifically horses #02, #03, #09, and #13. The analysis of sequence variations for the same horse during the same session also showed that five horses tended to change categories at the end of the session, specifically horses #03, #09, #13, and #15. Finally, the horses that had a consistent category usually warmed up using the same category, like horses #01 and #08. The horses that used multiple categories tended also to favor during warm-up the categories they would use predominantly during swimming, depending on the side, like horse #15.

### 3.6. Spatio-Temporal Parameters

The results for the spatio-temporal parameters defined in Section 2.3 (Figure 5b) in relation with the categories are presented Figure 10, while their means and standard deviations along with the results of the statistical analysis are compiled in Table 5. The range of duty factors for the front and hindlimbs was similar for CatA, CatB, and CatC, but they were significantly different (p<0.001) for CatD, with a reduced duty factor for this category (four beats with little overlap). Lateral delay and diagonal overlap exhibited differences between CatB and CatC, CatB and CatD, and CatC and CatD, while diagonal delay and lateral overlap were significantly different for CatA with respect to the other three categories, with, as expected, higher lateral overlap for CatA (two beats lateral) and higher diagonal overlap for CatB (two beats diagonal). Figure 10 and Table 5 illustrate that no single parameter showed significant differences across all category pairs.

## 4. Discussion

All horses in our studied population follow the same limb propulsion sequence: if starting with the left front limb, the next propulsive limb is the left hindlimb, then the right front limb, and finally the right hindlimb. However, this study demonstrates that the timing of these movements differs, with varying degrees of overlaps between these limbs. The veterinarians identified three temporal coordination patterns: strategy S1 as a lateral two-beat strategy characterized by the pairing of limbs in lateral fore–hind pairs, strategy S2 as a diagonal two-beat strategy characterized by the pairing of limbs in diagonal fore–hind pairs, and strategy S3 as a four-beat strategy characterized by the desynchronization of each limb. Qualitatively, for the two-beat strategies (strategies S1 and S2), it was hypothesized that the delay (lateral or diagonal) should be small with respect to duty factors (front or hind), meaning that the pairs of limbs (lateral or diagonal) will be synchronized, resulting in large overlaps (lateral or diagonal). In strategy S3, the four beats are regular and decomposed, so the delay should be large with respect to the duty factors, resulting in relatively small overlaps. By using indices (α and β) reflecting the ratio between these delays and the duty factor, it was therefore assumed that these different strategies could be automatically classified. The clustering performed by the machine learning algorithm with the quantitative parameters in the plane (α,β) confirms and refines the visual classification established by the veterinarians. This approach enables the computation of thresholds, allowing each horse’s laps to be classified into one of the three strategies. For the four-beat strategy, two subcategories were identified: one with a low overlap percentage and another with a medium overlap percentage. During the 12 laps performed by each horse, 5 horses maintained the same strategy, while 6 others changed their strategy over time, demonstrating inter-individual variability. The latter used two or three different swimming patterns, demonstrating intra-individual variability. Based on their observations, Grossi et al. [19] also identified three swimming styles. However, the names and categories they described differ from those in the present study. The descriptions used by Grossi et al. [19] for the three main styles associates what they call "connected pacer" with the two-beat swimming style with lateral overlap, and what they call “disconnected pacer” with the four-beat swimming style. However, it was impossible to classify horses in the “rear engine” category, as in our study, no horse was observed to swim using only the hindlimbs. Furthermore, the descriptions provided by Grossi et al. [19] for each of the subcategories were not detailed enough to facilitate a direct comparison with the results of our study.

All of the horses included in our study were able to move through the water, but they adopted different strategies. Over the past decades, the analysis of swimming in mammals has generated significant scientific interest, aiming to better understand their evolutionary adaptations to the aquatic environment and the limits of their swimming performances [26,27]. Researchers have naturally sought to compare these aquatic movements with well-known terrestrial gaits to establish links between limb coordination during terrestrial and aquatic locomotion [28,29,30,31,32]. It has been suggested that the neuromotor patterns controlling swimming likely evolved from pre-existing patterns used for walking. However, full immersion in water introduces biomechanical constraints that are distinct from those encountered on land, such as floatability and propulsion. In terrestrial locomotion, longer ground contact times are required to maintain stability, while buoyancy in water provides this stabilizing effect. This allows for aquatic limb movements that would be unstable on land [33], such as the bipedal swimming seen in small mammals (mice, rats, gerbils) using only their hindlimbs [8,31]. However, such bipedal swimming seems physically impossible for horses, due to their body structure, which would cause them to tilt vertically if passive, with the head remaining above water. In water, the horse’s center of gravity is located behind its center of buoyancy, resulting in a downward tilt of the body [34]. Dense masses such as the hindlimbs and abdomen tend to sink, whereas less-dense areas, like the thorax, tend to float. Consequently, a righting moment occurs, gradually bringing the horse—if it remains passive—into a position of equilibrium. In this state, the hindquarters sink to the bottom, aligning the center of gravity and the center of pressure along the same vertical axis. To counteract this, buoyancy must be actively managed through quadrupedal swimming; a technique commonly referred to as the “dog paddle” in large mammals, including horses [8,26,27].

Our results demonstrated variations in the horse’s limb coordination during swimming. Other study have also shown that there are variations in quadrupedal swimming within the same species, with each individual adopting the technique best suited to its morphology. For example, two species of squirrels, Citellus columbianus and Citellus franklini, display different swimming techniques; the former uses all four limbs, while the latter relies solely on its hindlimbs [8]. However, horses are morphologically closer to dogs compared to squirrels, suggesting potential similarities in their swimming techniques. Recent studies on dogs have shown that their limb coordination during swimming differs from classical terrestrial gaits, confirming that swimming requires specific adaptations [32]. In dogs, swimming involves a four-beat gait without simultaneous propulsive phase of the contralateral limbs, characterized by short propulsive phases and long return phases to optimize thrust and reduce drag in the water [35]. This sequence appears stereotyped in dogs. While one of the limb movement sequences we observed in some horses matches that of dogs, the variations seen in horses are not present in dogs. In summary, although similarities in limb movement patterns between horse and dog swimming might have been anticipated, our findings highlight variations in limb coordination in horses that have not been observed in other quadrupedal mammals such as dogs.

We observed significant variations in the swimming abilities of our horses, not only between individuals but also within the same individual. At this stage, it is interesting to consider how each horse selects its swimming strategy and what factors might influence this choice. Observing the horses’ warm-up strategies can provide valuable insights into their swimming preferences and habits, depending on the direction (hand) used. Generally, horses that maintain a consistent strategy tend to use that same approach during their warm-up. In contrast, horses that employ multiple strategies during swimming also tend to favor those strategies during warm-up, depending on the direction. Several explanations could account for these observations. Firstly, this study indicates that the direction in which the horse swims—whether it turns left or right in the semi-circular pool—appears to impact the swimming strategy used, as noted in laps on the left versus the right hand. The effect of negotiating these turns is a crucial factor to analyze. We observed that some horses adapted their movement to navigate the turns successfully by exerting more force with the limbs on the outside of the turn. Thus, when swimming on the left, horses may use their right lateral limbs more, and vice versa for the right. Since the turn occurs either before or after the cameras depending on the swimming direction, this could explain the variations observed in the strategies used between the two directions of rotation. The extent of these variations depends on the natural swimming abilities of each horse. It has been noted that, in most of the small mammals studied (except the guinea pig), young individuals tend to swim in circles during their first experiences [8]. An individual might swim to the right or left using all four limbs, with no inherent directional preference. The direction seemed to be influenced by other factors such as the initial orientation of the head. In our study, the horses were guided using two lunge lines, each held by a person on either side of the animal. It is possible that the forces exerted by these two handlers were not strictly equal, with the aim of helping the horse negotiate the turns [2]. The horse, having sensed the direction it would swim even before entering the pool, could anticipate a strategy to prepare for the upcoming turn. This difference in the forces exerted on the lunges, combined with the horse’s prior perception of the direction of rotation, could explain some of the variations observed in the swimming strategies employed to negotiate the turns [2].

Second, the training protocol included a break or a treadmill session, which appeared to influence the choice of swimming strategy when the horse returns to the aquatic environment. This effect might be attributed to the differences in sensory stimulation between the two environments. In the water, proprioceptive sensors and neurosensory receptors engaged differently. Thus, it can be assumed that an adaptation period is necessary for the horse to adjust to the new sensory information when first re-entering the aquatic environment [36]. For example, horse #09, after a treadmill session, initially adopted a two-beat strategy with lateral overlap upon returning to the pool before reverting to its usual four-beat strategy with medium overlap. This phenomenon of sensory re-adaptation during environmental changes is an important factor to consider when developing equine rehabilitation programs. Finally, other factors such as fatigue or level of training could also explain the variations observed in the swimming strategies adopted by the horses. It can be assumed that a fatigued horse, especially towards the end of a session, might modify its locomotor pattern to adopt a less energy-demanding strategy. Indeed, the muscles involved in keeping the head out of the water are heavily engaged [12]. Due to fatigue, the use of certain muscles may be limited to the bare minimum; just enough to keep the nose above water [1]. This explanation aligns with the observations made on some horses such as #03, #09, #13, and #15. These horses maintained a stable swimming strategy at the beginning of the session, but switched strategies towards the end (to a four-beat strategy—CatC or CatD), likely due to the onset of muscle fatigue. Thus, fatigue may prompt the horse to adopt a more energy-efficient swimming strategy to conserve its remaining energy resources. Therefore, this fatigue factor should be considered when analyzing strategy variations over a swimming session.

In this study, we applied machine learning algorithms, specifically k-means clustering, to evaluate the discriminative capability of two quantitative parameters for classifying limb coordination patterns in horse swimming. Qualitative observations made by veterinarians initially identified at least three distinct categories of swimming strategies. Using coordination graphs, we identified eight independent quantitative parameters for each lap across all horses, resulting in 125 observations. Analyzing these eight indicators alone and without prior knowledge made it difficult to discern clear trends across the different strategies. Machine learning proved advantageous in effectively handling large datasets [37]. However, managing eight indicators remains impractical for routine use. For effective use of k-means clustering, we reduce these variables to two ratios [38,39]. Although k-means clustering is an unsupervised machine learning technique, it provided an opportunity to explore different cluster numbers and thresholds. Initially, without prior knowledge, we applied a single k-means clustering with three clusters to identify strategies. While this approach provided some insights, it resulted in an overall accuracy of 77.6%, with 26 errors (20.8%), particularly between strategy S3 and category CatB. Additionally, there were minor misclassifications, such as incorrectly labeling strategy S1 as CatC or CatD (four-beat strategy), which were inconsistent with expected patterns. To improve the classification, we then implemented two successive k-means clusterings, each with two clusters. This refined approach yielded more consistent results for most laps, achieving an overall accuracy of 80.0%, with 24 errors (19.2%), again mainly between strategy S3 and category CatB. Encouragingly, both methods produced three clusters that corresponded to the strategies identified by the veterinarian team. However, the frequent mislabeling between strategy S3 and CatB led to the consideration of a fourth category. Acknowledging potential variations within the four-beat strategy, the second k-means clustering applied three clusters to establish quantitative thresholds, successfully revealing two distinct variations of the four-beat strategy. In doing so, the machine learning algorithm enhanced the qualitative observations made by the veterinarian team, providing quantitative insights that extended beyond visual identification. In our study, machine learning primarily served as a powerful tool to derive complex quantitative results in our study. A key strength of our approach was the iterative comparison and validation of findings with the observations of the veterinarian team. This iterative process is essential in studies employing machine learning algorithms, as it helps to bridge computational challenges with expert insights. However, as shown by our initial attempt using a single k-means clustering with three clusters, the application of machine learning to formulate hypotheses in medical or veterinary fields must be undertaken with caution [40,41,42].

This study has certain limitations. The main limitation is the relatively small sample size: eleven horses from a single breed, with twelve passages recorded, each containing only two or three cycles per passage, all recorded within a specific week. This constraint primarily stems from our recording methods [11], which rely on underwater cameras. The pre- and post-processing of these videos are time-consuming, limiting the amount of data we could feasibly collect. Current methods, such as Inertial Measurement Units (IMUs), could help address these challenges in the future by streamlining data collection. Nevertheless, video-based methods remain the only approach capable of providing detailed information about swimming strategies. Despite the small sample size, our study established a rational method to identify and automate the detection of different swimming strategies in horses and identified significant variations in their selection.

## 5. Conclusions

This study enabled a detailed analysis of the swimming strategies employed by horses during aquatic training sessions in a U-shaped pool. We observed inter- and intra-individual variability in limb coordination patterns during swimming. Three primary swimming strategies were identified and quantitatively characterized: the two-beat strategy with lateral overlap, the two-beat strategy with diagonal overlap, and the four-beat strategy with varying degrees of overlap, which could be further divided into two subcategories: the four-beat with medium overlap and the four-beat with little overlap. A classification system based on spatio-temporal indices allowed us to define objective and quantitative thresholds to distinguish between these strategies. While some horses maintained a consistent swimming strategy throughout, others exhibited an adaptive capacity by changing strategies within the same session. Several factors may influence this choice of strategy, including the swimming direction (right or left), the presence of turns in the pool, as well as fatigue. This variability highlights the complexity of aquatic locomotion in horses and questions the simplistic notion of “dog paddle” often used to describe swimming in terrestrial quadrupedal mammals. Our results suggest that horses demonstrate remarkable plasticity in adapting to the constraints of the aquatic environment. Further studies will be necessary to demonstrate potential cause-and-effect relationships between breed, usual terrestrial activity, direction of rotation, fatigue, or training, and the choice of swimming strategy by each horse. The present study lays the groundwork for a reproducible and automatable method to classify swimming strategies. To advance this further, it will now be necessary to develop simplified measurement methods (for instance, IMU-based recordings) to calculate the α and β indices in larger horse populations and under different circumstances. Deepening the understanding of the links between the strategies employed, swimming abilities, and the impact on the health and recovery of injured horses will aid in optimizing equine aquatic training programs in the near future.

## Figures and Tables

**Figure 1 animals-15-00195-f001:**
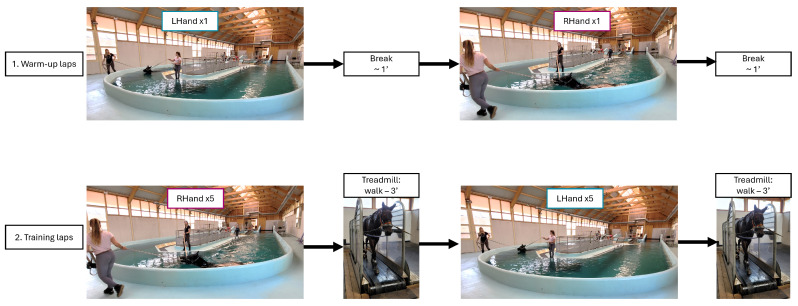
Description of the chronological structure of a typical swimming session (LHand: Left-Hand laps, RHand: Right-Hand laps, 1’: one minute, 3’: 3 min).

**Figure 2 animals-15-00195-f002:**
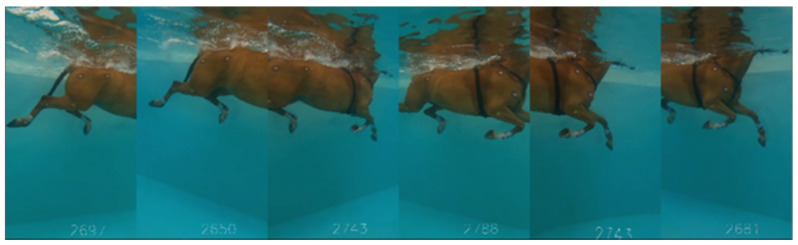
Assembled view from the 6 GoPro cameras during a right-hand lap of Horse #06.

**Figure 3 animals-15-00195-f003:**
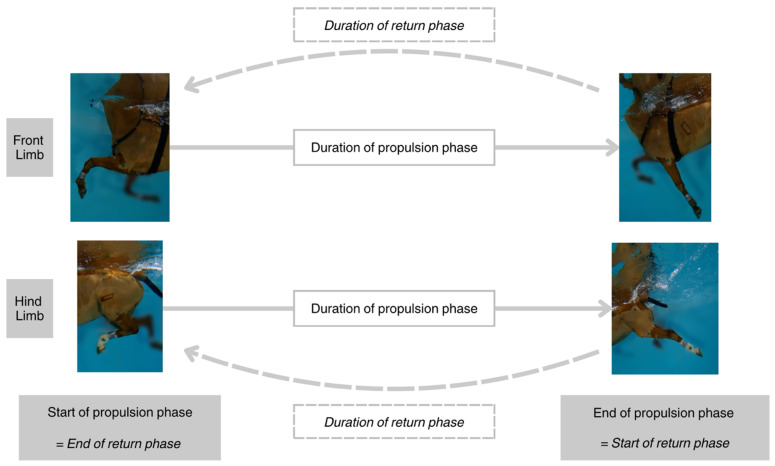
Duration of the propulsion (PP) and return phases (RP) of the forelimbs and hindlimbs.

**Figure 4 animals-15-00195-f004:**
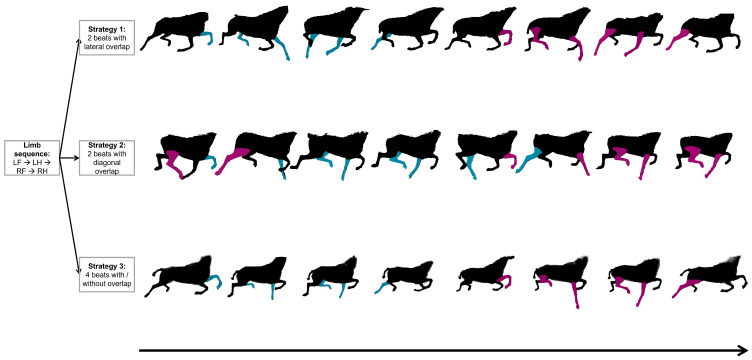
Identification of the different swimming strategies, beginning with the propulsion phase of the left front limb (LF = left forelimb; LH = left hindlimb; RF = right forelimb; RH = right hindlimb). The limbs are colored when they are in the propulsion phase; in blue for the left limbs and in pink for the right limbs.

**Figure 5 animals-15-00195-f005:**
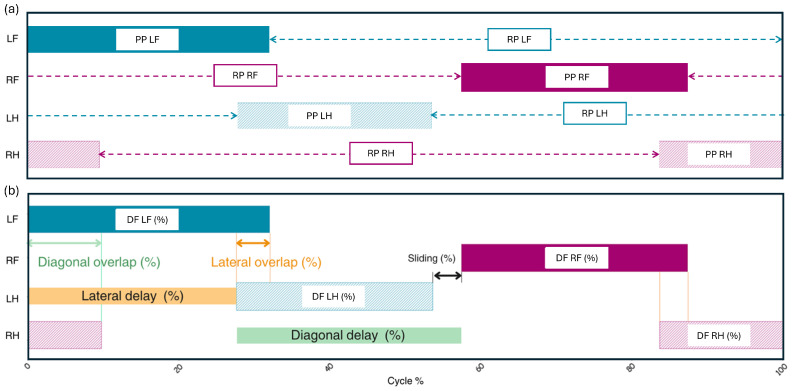
(**a**) Example (horse #02, lap #02) of the coordination sequence during the swimming cycle, with LF = left forelimb; LH = left hindlimb; RF = right forelimb; RH = right hindlimb. The propulsion phase (PP) is shown as the colored sections, while the return phase (RP) is left blank for each limb (pointed out with the arrows). (**b**) Definition of the quantitative parameters expressed as percentage of the cycle duration: Duty Factors (DF), diagonal overlaps, lateral overlaps, diagonal delay, lateral delay.

**Figure 6 animals-15-00195-f006:**
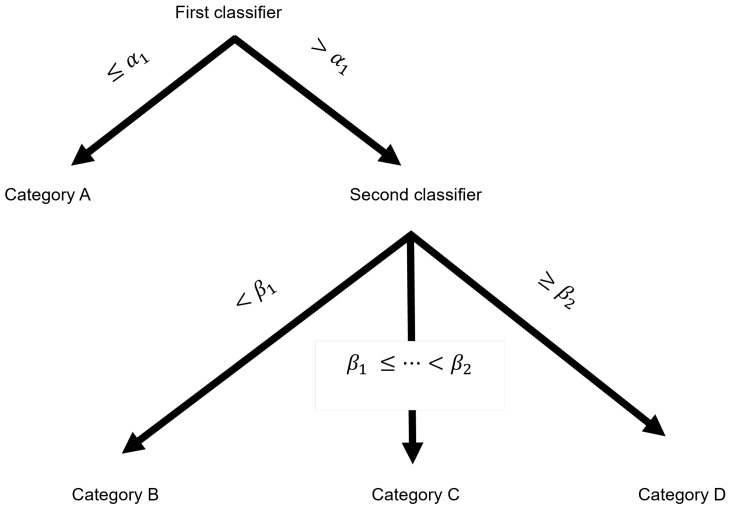
Description of the algorithm used to classify the swimming strategies. The first classifier gives the $\alpha_{1}$ threshold that separates CatA from the other categories. The second classifier separates the three lasting categories, CatB, CatC, and CatD, using $\beta_{1}$ and $\beta_{2}$ thresholds.

**Figure 7 animals-15-00195-f007:**
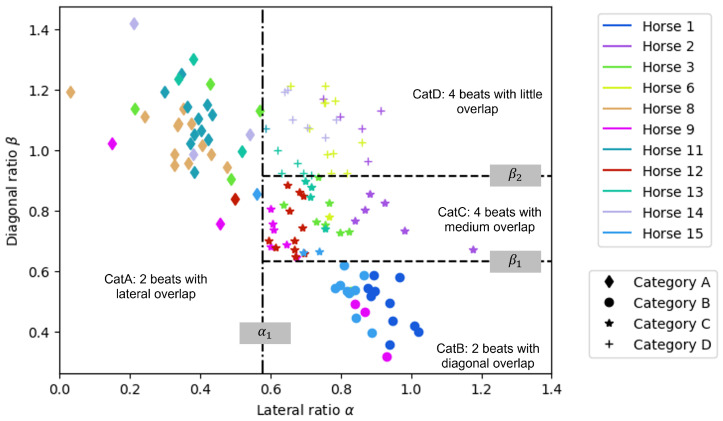
Results of the two-step classifier algorithm: 4 categories have been identified by the k-means algorithm (one color = one horse, and one symbol = one category), with threshold $\alpha_{1}$ separating CatA from the 3 others, and $\beta_{1}$ and $\beta_{2}$ separating CatB, CatC, and CatD.

**Figure 8 animals-15-00195-f008:**
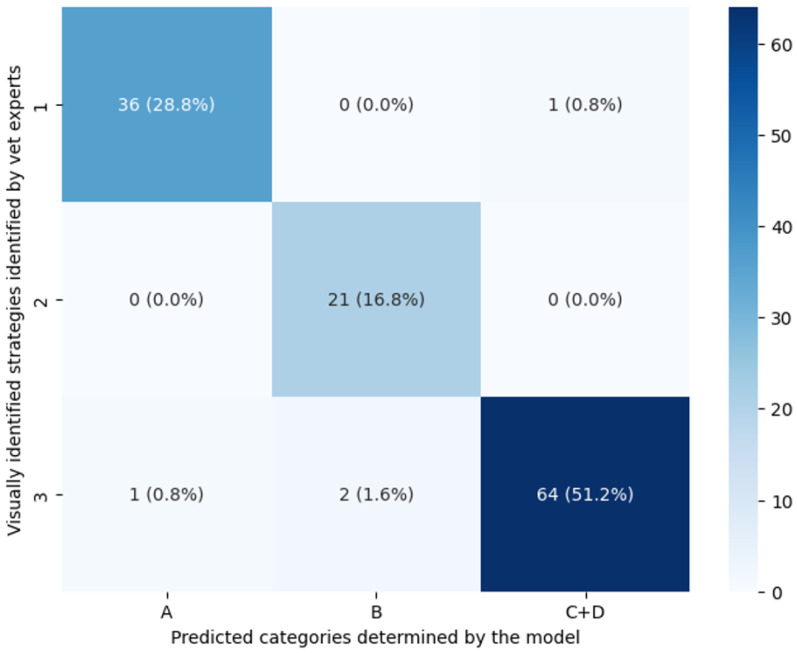
Confusion matrix between the strategies identified visually by the veterinarian experts (vet strategy) and the categories determined by the two-step classifier (predicted category) for all the laps considered (the percentage is relative to the total number of laps).

**Figure 9 animals-15-00195-f009:**
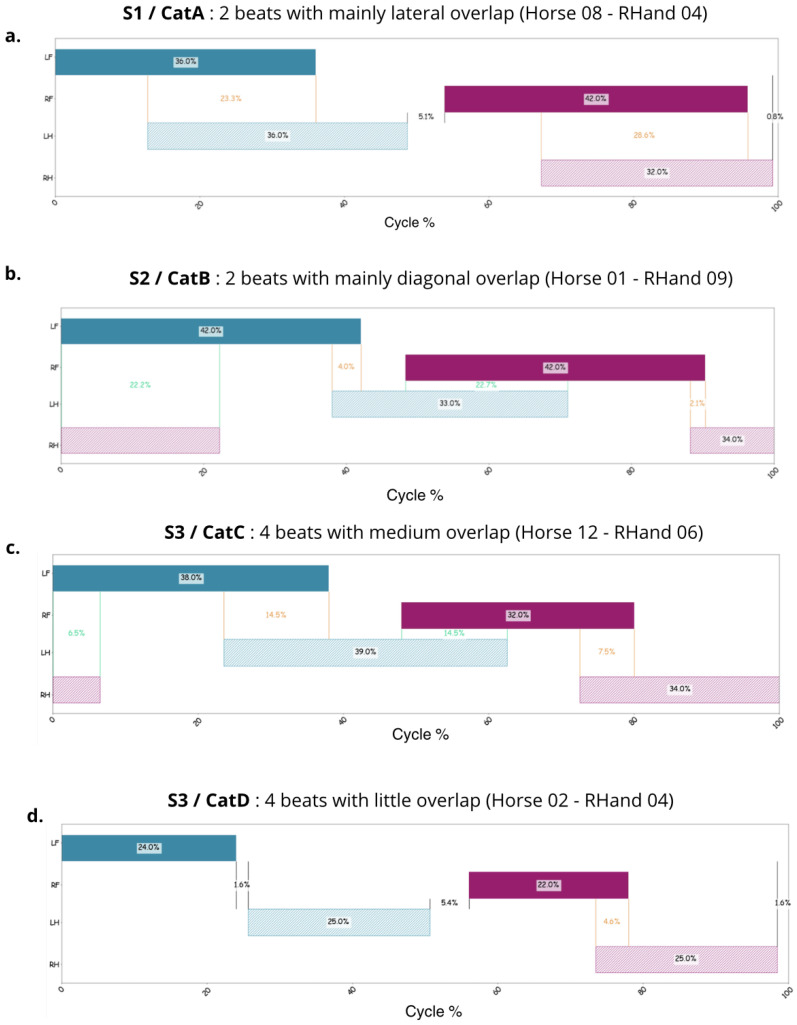
Correspondence between categories A, B, C, and D and strategies S1, S2, and S3. (**a**) Example of horse #08, fourth RHand: correspondence between category A (two beats with mainly lateral overlap) and strategy S1, (**b**) Example of horse #01, ninth RHand: correspondence between strategy S2 and category B (two beats with mainly diagonal overlap), (**c**) Example of horse #12, sixth RHand: correspondence between strategy S3 and category C (four beats with medium overlap), (**d**) Example of horse #02, fourth RHand: correspondence between strategy S3 and category D (four beats with low overlap).

**Figure 10 animals-15-00195-f010:**
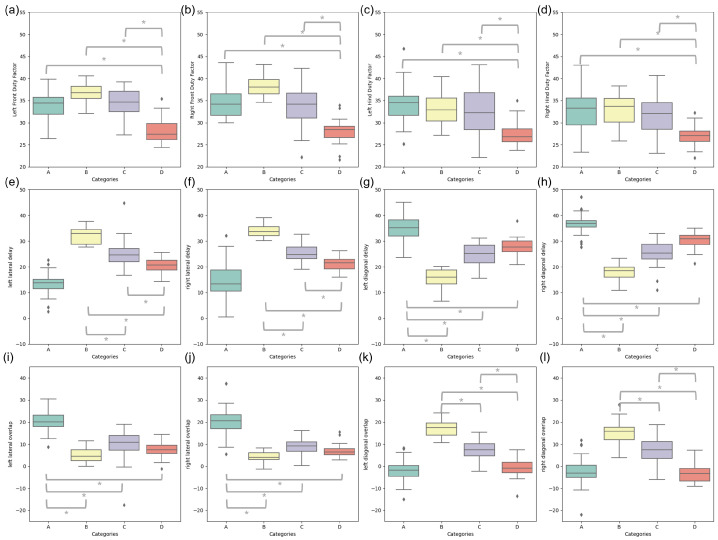
Boxplots of the different parameters (in %) from the coordination graph for the four categories (CatA—green, CatB—yellow, CatC—violet, and CatD—red) for (**a**) left front limb duty factor, (**b**) right front limb duty factor, (**c**) left hindlimb duty factor, (**d**) right hindlimb duty factor, (**e**) left lateral delay, (**f**) right lateral delay, (**g**) left diagonal delay, (**h**) right diagonal delay, (**i**) left lateral overlap, (**j**) right lateral overlap, (**k**) left diagonal overlap, (**l**) right diagonal overlap. Y-values are comprised between 20% and 55% for panel (**a**–**d**), between −10% and 50% for panel (**e**–**h**), and between −25% and 45% for panel (**i**–**l**). Two boxplots connected by a bridge and a ∗ symbol have statistically different means (p<0.001).

**Table 1 animals-15-00195-t001:** Description of the population of horses included in this study.

Horse Identification Number	Age (Years)	Gender (Mare/Gelding)	Height (cm)	Breed
#01	11	M	164	French warmblood
#02	14	M	160	French warmblood
#03	12	G	148	French pony
#06	11	G	166	French warmblood
#08	11	G	166	French warmblood
#09	12	G	170	French warmblood
#11	14	G	166	French warmblood
#12	12	G	170	French warmblood
#13	10	G	177	French warmblood
#14	9	G	177	French warmblood
#15	8	M	166	French warmblood

**Table 2 animals-15-00195-t002:** Number and percentage of laps identified as belonging to one of the three swimming strategies visually identified by expert veterinarians, for each horse individually (rows 1 to 11) and for all horses combined (last row). LHand corresponds to laps performed on the left, and RHand corresponds to laps performed on the right.

Horse Identification	Total Number of Laps Analysed	Strategy S1	Strategy S2	Strategy S3	Dominant Strategy
#01	10	0	LHand: 4 (40%)	0	S2 (100%)
			RHand: 6 (60%)		
#02	11	0	0	LHand: 4 (36%)	S3 (100%)
				RHand: 7 (64%)	
#03	12		0	LHand: 6 (50%)	S3 (58%)
		RHand: 5 (42%)		RHand: 1 (8%)	
#06	12	0	0	LHand: 6 (50%)	S3 (100%)
				RHand: 6 (50%)	
#08	12	LHand: 7 (58%)	0	0	S1 (100%)
		RHand: 5 (42%)			
#09	11	LHand: 3 (27%)		LHand: 2 (18%)	S1 (45%)
		RHand: 2 (18%)	RHand: 2 (18%)	RHand: 2 (18%)	
#11	12	LHand: 6 (50%)	0	0	S1 (100%)
		RHand: 6 (50%)			
#12	12		0	LHand: 6 (50%)	S3 (92%)
		RHand: 1 (8%)		RHand: 5 (42%)	
#13	12	LHand: 1 (8%)	0	LHand: 5 (42%)	S3 (75%)
		RHand: 2 (17%)		RHand: 4 (33%)	
#14	9	LHand: 1 (11%)	0	LHand: 5 (56%)	S3 (67%)
		RHand: 2 (22%)		RHand: 1 (11%)	
#15	12	LHand: 1 (8%)	LHand: 2 (17%)	LHand: 3 (25%)	S2 (67%)
			RHand: 6 (50%)		
Total	125	LHand: 18 (14%)	LHand: 6 (5%)	LHand: 37 (30%)	S3 (51%)
		RHand: 24 (19%)	RHand: 14 (11%)	RHand: 26 (21%)	

**Table 3 animals-15-00195-t003:** Number and percentage of laps identified as belonging to one of the four swimming categories identified by the two-step classifier, for each horse individually (rows 1 to 11) and for all horses combined (last row). LHand corresponds to laps performed on the left, and RHand corresponds to laps performed on the right.

Horse Identification	Total Number of Laps Analyzed	Category A	Category B	Category C	Category D	Dominant Category
#01	10	0	LHand: 4 (40%)	0	0	catB (100%)
			RHand: 6 (60%)			
#02	11	0	0	LHand: 2 (18%)	LHand: 2 (18%)	catC (55%)
				RHand: 4 (37%)	RHand: 3 (27%)	
#03	12		0	LHand: 5 (42%)	LHand: 1 (8%)	catC (50%)
		RHand: 5 (42%)		RHand: 1 (8%)		
#06	12	0	0	LHand: 1 (8%)	LHand: 5 (42%)	catD (59%)
				RHand: 4 (33%)	RHand: 2 (17%)	
#08	12	LHand: 7 (58%)	0	0	0	catA (100%)
		RHand: 5 (42%)				
#09	11	LHand: 2 (18%)		LHand: 3 (27%)	0	catC (45%)
		RHand: 1 (10%)	RHand: 3 (27%)	RHand: 2 (18%)		
#11	12	LHand: 6 (50%)	0	0	0	catA (100%)
		RHand: 6 (50%)				
#12	12		0	LHand: 6 (50%)	0	catC (92%)
		RHand: 1 (8%)		RHand: 5 (42%)		
#13	12	LHand: 2 (17%)	0	LHand: 4 (33%)	0	catC (66%)
		RHand: 2 (17%)		RHand: 4 (33%)		
#14	9	LHand: 1 (11%)	0	0	LHand: 5 (56%)	catD (67%)
		RHand: 2 (22%)			RHand: 1 (11%)	
#15	12	LHand: 1 (8%)	LHand: 3 (25%)	LHand: 2 (17%)	0	catB (75%)
			RHand: 6 (50%)			
Total	125	LHand: 19 (15.2%)	LHand: 7 (5.6%)	LHand: 23 (18.4%)	LHand: 13 (10.4%)	
		RHand: 22 (17.6%)	RHand: 15 (12.0%)	RHand: 20 (16.0%)	RHand: 6 (4.8%)	

**Table 4 animals-15-00195-t004:** Results for the category-wise metrics.

Categories	Precision	Recall	F1 Score	Support (Number of Laps)
A	0.97	0.97	0.97	37
B	0.91	1.00	0.95	21
C + D	0.98	0.96	0.97	67

**Table 5 animals-15-00195-t005:** Mean and standard deviation of the quantitative parameters with respect to categories A to D. Significant differences are indicated by the numbers in parenthesis following the name of the parameter. Category pairs that are significantly different (with a *p* value < 0.001) for the parameter considered: (1) Cat A and CatD, CatB and CatD, CatC and CatD, (2) CatB and CatC, CatB and CatD, CatC and CatD, (3) CatA and CatB, CatA and CatC, CatA and CatD.

	CatA		CatB		CatC		CatD	
	Left	Right	Left	Right	Left	Right	Left	Right
Forelimb duty factor (1)	33.6±3.3%	34.6±3.2%	36.9±2.2%	38.2±2.3%	34.6±2.9%	33.9±4.5%	28.2±2.8%	28.0±2.7%
Hindlimb duty factor (1)	34.3±3.9%	33.2±4.5%	33.5±3.7%	33.0±3.5%	32.6±5.2%	31.9±4.2%	27.4±2.7%	27.1±2.3%
Lateral delay (2)	13.4±4.0%	14.8±6.5%	32.2±3.2%	34.2±2.4%	24.8±5.2%	25.4±3.3%	20.6±2.9%	21.1±2.6%
Diagonal delay (3)	35.0±4.5%	36.8±3.7%	15.8±3.4%	17.8±3.3%	24.6±4.4%	25.3±4.4%	27.9±3.3%	30.4±3.2%
Lateral overlap (3)	20.2±4.6%	19.9±6.1%	4.7±3.1%	4.0±2.6%	9.8±6.9%	8.6±3.3%	7.6±3.5%	7.0±3.0%
Diagonal overlap (2)	−1.8±5.4%	−2.4±6.0%	17.2±3.5%	15.7±5.3%	7.3±4.2%	7.3±6.4%	−0.8±4.1%	−3.0±4.5%

## Data Availability

The data that support the findings of this study are available from the corresponding author upon reasonable request.

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
