# Peer review of "Description and Analysis of Horse Swimming Strategies in a U-Shaped Pool"

_animals, 2025, doi:10.3390/ani15020195_

Round 1

Reviewer 1 Report

Comments and Suggestions for Authors

I found this an interesting paper, the first I have seen describing swimming gait in detail, using an innovative and elegant technique. I have only minor comments:

1. Line 12 and 30: Check simple summary and abstract for consistency of use of swimming for training and/or rehabilitation. I believe it is used for both, so both summary and abstract should reflect this. Full stop required line 30.

2. Lines 37-41: What is the evidence for improving rehabilitation outcomes and for engaging in weight bearing exercise sooner than otherwise?

3. Line 54 - what is 'joint amplitude'?

4. Line 63: Are you sure you mean 'respiratory capacity'? and not 'mechanics of breathing'? Capacity to me equates to either lung volume change or an improvement in function?

5. Was ethical approval not required? How were horses recruited? I see some had lesions not incompatible with pool exercise. Please expand on your recruitment/inclusion criteria and the 4 excluded. One had colic - was this post swim?

6. Figure 1: define 1' and 3' in legend.

7. Figure 2 shows the horses wearing equipment, a roller, breastplate (?) and boots. In my experience this can affect gait. What headgear/bit did the horses wear? Please explain how the horses were acclimated to swimming in the equipment, or did they wear it from their first experience of swimming? 

8. Do you have any measure of swim speed?. I wonder if horses were encouraged at all to maintain swim speed or allowed to go at self selected speed?

9. Please give details of pool depth and water temperature. Depth purely for interest for others, and temperature because it affects viscosity and potentially, physiological response to swimming.

10. Line 211 - is left diagonal LF, RH?

11. figure 7 has all 15 horses in it, but I thought 4,5,7 and 10 were excluded before the 8th week and the measurements were taken in 8th week?

12. Table 5 legend: Significant not significative

Line 405: Re the centre of buoyancy being in front of the CoM producing a downward tilt. What is meant by downward? The centre of buoyancy is the centre of mass of the displaced water - surely this is behind the CoM of the horse, not in front? Please check and/or provide more detail. I did check out the reference but this did not give sufficient explanation for me either.

Line 448: I would say "two lunge lines" or "two lunge reins" rather than 2 lunges.

Author Response

1. Summary

Thank you very much for taking the time to review this manuscript and for the very specific comments which have helped us refine and improve our manuscript. Please find below the detailed responses and the corresponding revisions/corrections highlighted in the track changes in the resubmitted files. The line numbers in this document refer to the new submitted version of the manuscript.

We believe these revisions address the concern raised and contribute to enhancing the overall quality and clarity of the manuscript. Should further clarification or additional changes be required, we would be happy to address them promptly.

Thank you again for your valuable feedback.

2. Point-by-point response to Comments and Suggestions for Authors

Comments 1: Line 12 and 30: Check simple summary and abstract for consistency of use of swimming for training and/or rehabilitation. I believe it is used for both, so both summary and abstract should reflect this. Full stop required line 30.

Response 1: Thank you for highlighting this point. We fully agree with your comment and have addressed it accordingly. Specifically, we have modified the sentence on line 12 to read:

“Aquatic training has been integrated into equine rehabilitation and training programs for several decades.”

Additionally, we have added the point line 30.

Comments 2: Lines 37-41: What is the evidence for improving rehabilitation outcomes and for engaging in weight bearing exercise sooner than otherwise?

Response 2: Thank you for your comment. While direct evidence in horses is limited, we base our reasoning on relevant findings from animal models and comparative physiology. Studies have shown that total immobilization can lead to significant adverse effects such as muscle atrophy and reduced bone quality and density (Delguste et al., 2007). Research conducted on osteoporotic rats has demonstrated that swimming activity can, within three weeks, counteract the deleterious effects of inactivity on bone quality and even enhance it. Specifically, Falcai et al. (2015) reported significant improvements in bone mass (43%), bone strength (29%), trabecular thickness (58%), bone volume (85%), trabecular number (27%), and cortical volume (30%) compared to immobilized controls.

We have incorporated these findings into the manuscript, lines 37-41, with the following revisions:
The buoyancy of water reduces the impact and stress on vulnerable structures, allowing earlier engagement in weight-bearing activities compared to rehabilitation programs without swimming (Falcai et al., 2015). This approach has the potential to accelerate recovery, maintain muscle strength and joint mobility, and prevent the adverse effects associated with prolonged immobilization (Delguste et al., 2007).

The following references have been added to the manuscript:

  • Delguste, C., Amory, H., Doucet, M., et al. (2007). Pharmacological effects of tiludronate in horses after long-term immobilization. Bone, 41(3), 414–421. [https://doi.org/10.1016/j.bone.2007.05.005]
  • Falcai, M.J., Zamarioli, A., Leoni, G.B., et al. (2015). Swimming Activity Prevents the Unloading Induced Loss of Bone Mass, Architecture, and Strength in Rats. BioMed Research International, 2015, 507848. [https://doi.org/10.1155/2015/507848]

Comments 3: Line 54 - what is 'joint amplitude'?

Response 3: In our manuscript, when we refer to joint amplitude, we are specifically discussing the Range of Motion (ROM). ROM is defined as the degree of movement available at a joint, calculated by subtracting the maximum flexion angle from the maximum extension angle.

Thank you for bringing this to our attention. Upon reviewing our terminology, we acknowledge that the use of the term "amplitude" in our manuscript was not entirely appropriate. This was due to an error in translation, as "amplitude" is commonly used in French to describe joint mobility, whereas the correct term in English is "range of motion" (ROM).

To address this, we have replaced all instances of "amplitude" with "range of motion" throughout the manuscript to ensure accuracy and consistency with the standard terminology used in scientific literature.

We appreciate your feedback, which has allowed us to improve the precision of our language.

Comments 4: Line 63: Are you sure you mean 'respiratory capacity'? and not 'mechanics of breathing'? Capacity to me equates to either lung volume change or an improvement in function?

Response 4: Thank you for your insightful comment. We agree that in this context, the term "mechanics of breathing" is indeed more appropriate than "respiratory capacity." The latter could imply changes in lung volume or an improvement in overall respiratory function, which was not the intended meaning.

To clarify, studies focusing on respiratory parameters during swimming have evaluated variables such as respiratory rates, intratracheal pressures, inspiratory and expiratory times, respiratory cycles, oxygen consumption, inspired and expired volumes, peak inspiratory and expiratory flows, minute ventilation, and inspiratory-to-expiratory durations and ratios (Hobo et al., 1998; Jones et al., 2020; Leguillette et al., 2024). These measurements are better described as mechanics of breathing rather than capacity.

Accordingly, we have modified the sentence on line 65 to read:

Notable adaptations include changes in cardiovascular function [1,11,12], mechanics of breathing [13–15], and metabolic processes [1,12]...

We appreciate your careful review, which has helped us improve the accuracy of our terminology.

References:

Hobo, S.; Yoshida, K.; Yoshihara, T. Characteristics of Respiratory Function during Swimming Exercise in Thoroughbreds. J. Vet. Med. Sci. 1998, 60, 687–689. [https://doi.org/10.1292/jvms.60.687]

Jones, S.; Franklin, S.; Martin, C.; Steel, C. Complete Upper Airway Collapse and Apnoea during Tethered Swimming in Horses. Equine Vet. J. 2020, 52, 352–358.

Leguillette, R.; McCrae, P.; Massie, S.; Filho, S.A.; Bayly, W.; David, F. Workload and Spirometry Associated with Untethered Swimming in Horses. BMC Vet Res. 2024, 20, 327. [https://doi.org/10.1186/s12917-024-04143-3]

Comments 5: Was ethical approval not required? How were horses recruited? I see some had lesions not incompatible with pool exercise. Please expand on your recruitment/inclusion criteria and the 4 excluded. One had colic - was this post swim?
Response 5: Thank you for your comment. This study received ethical approval from the ComERC of the École Nationale Vétérinaire d’Alfort on 14/11/2022 (file number: 2022-09-19).

Each owner of the horses included in the study provided informed consent by signing a detailed consent form outlining the study protocol.

Regarding the inclusion criteria:

Horses were recruited through a public call for applications.

Eligibility criteria included:

  • Horses actively competing in show jumping or eventing in the previous year.
  • Aged between 5 and 17 years at the time of inclusion.
  • A history of sports-related issues linked to axial lesions.

The minimum inclusion criterion was the presence of cervical and/or dorsal radiographic abnormalities. Horses presenting with concurrent limb-related problems were excluded. Additionally, horses were excluded from the protocol under the following conditions:

·       If they failed to enter the pool after three habituation sessions.

·       If a lameness-related condition appeared during the first month of the program. This initial month constituted a terrestrial training phase aimed at conditioning the horses before introducing aquatic training, which began at the end of the first month.

Upon arrival, selected horses underwent a thorough examination, which included:

  1. Physical examination testing axial mobility, sensitivity, and morphometry.
  2. Standardized locomotor examination in-hand, ridden exercise including jumps, and on a high-speed treadmill.
  3. Imaging assessments (radiographs and ultrasounds).
  4. Comprehensive medical examinations including blood tests, a track exercise test, and echocardiography.

Regarding the four excluded horses:

  • Horse #04: Developed post swim colic during the 5th week of the protocol, coinciding with the introduction of aquatic work. Due to the need for hospitalization, it was excluded from the study.
  • Horse #05: Exhibited an adverse reaction to the glue used to attach the inertial measurement units (IMUs).
  • Horse #07: Became unsafe to handle during routine care, necessitating exclusion.
  • Horse #10: Displayed lameness during the third week and was excluded for this reason.

To provide additional precision and transparency for the readers, we propose adding the detailed inclusion criteria and information about the excluded horses to the appendix section (appendix A in the revised version of the manuscript). We believe this approach will enhance the accessibility of the information without overloading the main manuscript text.

We have also modified the sentence on lines 144–146 to clarify this point:

“[Eleven non-lame amateur show jumping horses, without limb injuries but potentially exhibiting cervical or dorsal lesions compatible with pool exercise, were included in the study (see Appendix A for details of the inclusion process and excluded horses).]”

We appreciate your suggestion and are happy to implement this change.

Comments 6: Figure 1: define 1' and 3' in legend.
Response 6: Thank you for your comment. We have updated the manuscript to include the following clarification:
“[Figure 1. Description of the chronological structure of a typical swimming session (LHand: Left-Hand laps, RHand: Right-Hand laps, 1’: one minute, 3’: three minutes).]”

We hope this modification provides additional clarity.

Comments 7: Figure 2 shows the horses wearing equipment, a roller, breastplate (?) and boots. In my experience this can affect gait. What headgear/bit did the horses wear? Please explain how the horses were acclimated to swimming in the equipment, or did they wear it from their first experience of swimming?
Response 7: All horses were equipped in the same manner throughout the study. This included:

·       A girth to hold an Equimeter.

·       A hunting breastplate to prevent the girth from slipping during exercises.

·       Inertial Measurement Units (IMUs) attached with adhesive tape (one on each limb, two on the back, and one at the poll).

·       A bonnet with ear covers.

·       A halter topped with a cavesson.

The horses wore this equipment from their very first swimming session and also during the ground training conducted in the first four weeks of the program. This setup ensured that the horses were fully acclimated to the equipment before data collection began. For this reason, pool measurements were only taken starting in the 5th week, after a sufficient adaptation period to both the equipment and the swimming environment.

We have modified the sentence on lines 161–162 to clarify this point:

“[After a three-week adaptation period (weeks 5 to 7, with nine swimming sessions, during which the horses wore monitoring equipment), the measurements were taken during the 8th week of training.]”

Comments 8: Do you have any measure of swim speed? I wonder if horses were encouraged at all to maintain swim speed or allowed to go at self selected speed?

Response 8: A study utilizing 3D underwater motion capture was conducted on six of our horses (#01, #02, #03, #06, #08, #09) in parallel of our protocol. This study, carried out by Moiroud et al. (2024) as part of the CAPT-ESE project, demonstrated that one complete swimming cycle lasted an average of 1.4 ± 0.1 seconds. This duration was derived by analyzing the propulsion phases (PP) and recovery phases (RP).

In our study, the horses were deliberately allowed to swim at their self-selected speeds to ensure natural and undisturbed swimming behavior during data collection. Moreover, it is inherently difficult to compel horses to swim faster or slower in the pool, as they instinctively regulate their pace based on their comfort and physical capability.

Future studies within our protocol will investigate the effects of training and the strategies choice on swimming efficiency, evaluated by speed relative to the number of cycles required.

We add the sentence on line 159 to clarify this point: “[A swimming session consisted of laps, which included entering the pool, swimming in the U-shape pool, and exiting the pool (Figure 1). The horses were deliberately allowed to swim at their self-selected speeds to ensure natural and undisturbed swimming behavior during data collection.]”

Reference:

Moiroud, C., Giraudet, C., Beaumont, A., et al. (2024). Determination of the different phases of the horse swim cycle and correlation to the limbs’ kinematics obtained by underwater 3D motion capture analysis: preliminary results. 5th Scientific Meeting of the European College of Veterinary Sports Medicine and Rehabilitation, Cordoba, Spain.

Comments 9: Please give details of pool depth and water temperature. Depth purely for interest for others, and temperature because it affects viscosity and potentially, physiological response to swimming.

Response 9: The U-shaped pool is 50 meters long and 3 meters deep, with independent entry and exit points, featuring an angle of approximately 15 degrees. The water temperature ranges between 15 and 22°C depending on the season.

We have updated the sentence on page 4, lines 142-144, to clarify this information:

“[The study was conducted in a U-shaped pool, 50 meters long and 3 meters deep, located at CIRALE (EnvA, Goustranville, France). The water temperature ranges between 15 and 22°C depending on the season.]” 

Comments 10: Line 211 - is left diagonal LF, RH?
Response 10: Left diagonal and right diagonal have their usual definition so the left diagonal corresponds to LF – RH and the right diagonal to RF – LH.
Accordingly, we have modified the sentence on lines 218- 219 to read: “[Lateral overlap (latOL) examines two ipsilateral fore and hind limbs, with latOLleft for the left limbs and latOLright for the right limbs. Diagonal overlap (DiagOL) examines two diagonal fore and hind limbs, with DiagOLleft for the left (LF-RH) and DiagOLright for the right diagonal (RF-LH).]”

Comments 11: figure 7 has all 15 horses in it, but I thought 4,5,7 and 10 were excluded before the 8th week and the measurements were taken in 8th week?

Response 11: Yes, that is correct; horses 4, 5, 7, and 10 do not appear in figure 7 because they were excluded before the eighth week. The results presented in this figure pertain solely to the 11 horses that remained in the study during week 8, with each horse represented by a unique color, resulting in 11 distinct colors.

We opted not to rename the horses after excluding the four from the protocol. This decision explains the numbering gaps, such as the transition from horse 3 to horse 6, from horse 6 to horse 8, and from horse 9 to horse 11, as well as the presence of a horse labeled as horse 15.

We hope this clarification resolves any confusion.

Comments 12: Table 5 legend: Significant not significative

Response 12: Thank you, we agree. We modified the legend of table 5: “[Mean and Standard deviation of the quantitative parameters with respect to the categories A to D. The significant difference is indicated by the number in parenthesis following the name of the parameter. Couple of categories that are significantly different (with a pvalue < 0.001) for the parameter considered: (1) Cat A & CatD, CatB & CatD, CatC & CatD, (2) CatB & CatC, CatB & CatD, CatC & CatD, (3) CatA & CatB, CatA & CatC, CatA & CatD.]”

Comments 13: Line 405: Re the centre of buoyancy being in front of the CoM producing a downward tilt. What is meant by downward? The centre of buoyancy is the centre of mass of the displaced water - surely this is behind the CoM of the horse, not in front? Please check and/or provide more detail. I did check out the reference but this did not give sufficient explanation for me either.

Response 13: While it may not seem intuitive, we observed that when a horse stops swimming, its body naturally adopts a more vertical position, with the head remaining above water.

Dense masses, such as the hind limbs and abdomen tend to sink, whereas less dense areas, like the thorax, tend to float. Consequently, a righting moment occurs, gradually bringing the horse—if it remains passive—into a position of equilibrium. In this state, the hindquarters sink to the bottom, aligning the center of gravity and the center of pressure along the same vertical axis.

We hope this explanation addresses your query.

Comments 14: Line 448: I would say "two lunge lines" or "two lunge reins" rather than 2 lunges.

Response 14: Thank you for your suggestion; it is indeed clearer. We have updated the sentence on page 21, line 455, to read: “[In our study, the horses were guided using two lunge lines, each held by a person on either side of the animal.]”

Reviewer 2 Report

Comments and Suggestions for Authors

A few comments:

Line 142:  Why are horses with cervical or dorsal lesions included? (versus clinically normal horses with no limb or spinal lesions?)

Line 160:  Why were the 2 horses different than the others?

Author Response

 1. Summary

Thank you very much for taking the time to review this manuscript and to help us improve its quality. Please find below the detailed responses and the corresponding revisions/corrections highlighted in the track changes in the resubmitted files. The line numbers in this document refer to the new submitted version of the manuscript.

We believe these revisions address the concern raised and contribute to enhancing the overall quality and clarity of the manuscript. Should further clarification or additional changes be required, we would be happy to address them promptly.

Thank you again for your valuable feedback.

2. Point-by-point response to Comments and Suggestions for Authors

Comments 1: Line 142:  Why are horses with cervical or dorsal lesions included? (versus clinically normal horses with no limb or spinal lesions?)

Response 1: Thank you for pointing this out. The CAPT-ESE project, within which this study is conducted, seeks to investigate several hypotheses. One of these is whether working in an aquatic environment can enhance the clinical tolerance of cervical or dorsal spinal lesions. For this reason, the horses included in this methodological study also had radiographic abnormalities in the cervical and thoracolumbar spine. However, the inclusion criteria ensured from the beginning of the protocol that all of these horses were able to swim naturally and without hesitation.

Comments 2: Line 160:  Why were the 2 horses different than the others?

Response 2: Horses #01 and #02 were the first to be captured on video. During their initial motion capture session, we encountered technical issues, prompting us to repeat the experiment the following day while adhering to the swimming protocol of their next scheduled session.

The technical issue did not impact the results regarding their swimming strategies. The analysis of these strategies, including the direction of the laps and treadmill pauses, was adjusted to align with their training protocol for that session.
